# Healthcare usage and expenditure among people with type 2 diabetes and/or hypertension in Cambodia: results from a cross-sectional survey

Srean Chhim [1,2] Vannarath Te [1,2,3,4] Veerle Buffel [5] Josefien van Olmen,[4] Savina Chham,[1,6] Sereyraksmey Long,[2] Sokunthea Yem,[1] Wim Van Damme,[7,8] Edwin Wouters [5] Ir Por[9]

For numbered affiliations see end of article.

**Correspondence to**
Srean Chhim;
srean168@gmail.com

## ABSTRACT

**Objective** To assess usage of public and private healthcare, related healthcare expenditure, and associated factors for people with type 2 diabetes (T2D) and/or hypertension (HTN) and for people without those conditions in Cambodia.

**Methods** A cross-sectional household survey.

**Settings** Five operational districts (ODs) in Cambodia.

**Participants** Data were from 2360 participants aged ≥40 years who had used healthcare services at least once in the 3 months preceding the survey.

**Primary and secondary outcome** The main variables of interest were the number of healthcare visits and healthcare expenditure in the last 3 months.

**Results** The majority of healthcare visits took place in the private sector. Only 22.0% of healthcare visits took place in public healthcare facilities: 21.7% in people with HTN, 37.2% in people with T2D, 34.7% in people with T2D plus HTN and 18.9% in people without the two conditions (p value <0.01). For people with T2D and/or HTN, increased public healthcare use was significantly associated with Health Equity Fund (HEF) membership and living in ODs with *community-based care*. Furthermore, significant healthcare expenditure reduction was associated with HEF membership and using public healthcare facilities in these populations.

**Conclusion** Overall public healthcare usage was relatively low; however, it was higher in people with chronic conditions. HEF membership and *community-based care* contributed to higher public healthcare usage among people with chronic conditions. Using public healthcare services, regardless of HEF status reduced healthcare expenditure, but the reduction in spending was more noticeable in people with HEF membership. To protect people with T2D and/or HTN from financial risk and move towards the direction of universal health coverage, the public healthcare system should further improve care quality and expand social health protection. Future research should link healthcare use and expenditure across different healthcare models to actual treatment outcomes to denote areas for further investment.

## INTRODUCTION

Type 2 diabetes (T2D) and hypertension (HTN) are global public health concerns.

## STRENGTHS AND LIMITATIONS OF THIS STUDY

⇒ Our study is among the few to examine healthcare usage and expenditure among people with type 2 diabetes (T2D) and/or hypertension (HTN) in comparison to people without these two conditions in Cambodia.

⇒ The sampling design—randomising villages, households and household members—is robust within its scope, targeting the population in rural or semi-rural settings in Cambodia.

⇒ The data collection is robust and ensures a reliable dataset.

⇒ The fact that the five operational districts were selected purposively limited the generalisability for the national level as most of the study sites we selected were rural or semi-rural.

⇒ The sample size for the group of people with T2D only and people with T2D plus HTN may be relatively small and may have insufficient power to assess the association between outcome variables and the dependent variables.

They are major risk factors for cardiovascular diseases, causing about 31% (17.9 million) of all deaths worldwide annually.[1] The prevalence of people with T2D and/or HTN will likely continue to increase.[2 3] These two diseases disproportionately affect low-income and middle-income countries and account for around 75% of all deaths in these countries.[1]

In Cambodia, large-scale population-based studies such as STEPS Surveys have shed light on the prevalence and risk factors of chronic diseases. The prevalence of T2D and HTN rose noticeably over recent years from 2.9% and 11.2% in the population aged 25–64 years in 2010 to 9.6% and 14.2% in the population aged 18–69 years in 2016, respectively.[4]

Previous nationally representative surveys have shown that a majority of the population seeks outpatient curative care in private

facilities, but knowledge on the related healthcare uptake and expenditure among those with T2D and/or HTN is scarce.[5] Additionally, these surveys do not cover topics related to the management of the diseases, such as healthcare usage and expenditure.[6] There is only one study, by Bigdeli *et al*, which examines access to care for people with T2D and/or HTN concerning social health protection schemes in Cambodia.[7] This study shows that 61% of the people with T2D and/or HTN who knew their status were diagnosed in private facilities.[7] However, this study collected data in 2013, before key interventions were introduced or expanded in public healthcare facilities.[8] Also, it provides limited information about which types of health facilities were used, and what differences exist between people with one or both conditions compared with those without.

In the last decade, the Cambodian Ministry of Health (MoH), in collaboration with development partners, made significant efforts to improve the quality of public healthcare and initiated a few healthcare delivery models for people with T2D and/or HTN in public facilities.[8] These models include *hospital-based care, health centre-based care, community-based care* and a combination of all three models (*coexisting care*) (box 1). These efforts might have changed the pattern of healthcare usage and related expenditure, especially among people with T2D and/or HTN.

A better understanding of the current patterns of healthcare usage and expenditure among people with T2D and/or HTN is critical for better resource allocations and strategies to improve the management of T2D and HTN. The main objectives of this study are twofold. First, it evaluates usage and determines the factors associated with public healthcare use in four groups: (1) people without T2D or HTN, (2) people with T2D alone, (3) people with HTN alone and (4) people with T2D plus HTN. Second, it assesses the healthcare expenditure in the 3 months preceding the survey for all services used by the four patient groups in public and private facilities and determines factors associated with (reducing or increasing) healthcare expenditure.

## Context

The health system in Cambodia is pluralistic, meaning healthcare services are provided by both public and private healthcare providers.[5 9]

*Public healthcare services* in Cambodia dominate preventive services (reproductive, maternal, neonatal and child health), control of primary disease (tuberculosis, malaria and HIV/AIDS control) and inpatient treatment.[5] The facilities include health posts, health centres, district referral hospitals, provincial referral hospitals and national hospitals.[5] Public healthcare is organised per operational district (OD)—the third and last administrative level in Cambodia's health system management.[5] An OD covers a population of 100 000–200 000 people while a health centre covers a population of 10 000–20 000 people.[5] Remote areas with a small population can

---

**Box 1  Overview of different care models in Cambodia in 2021**

⇒ The *hospital-based care* model is standard care, which means an operational district (OD) has a government-run non-communicable disease clinic at the district referral hospitals.[8] By 2021, 31 out of 117 referral hospitals had implemented this model. The patients refer themselves to the units (and they are thus not transferred by an intermediary unit).

⇒ The *health centre-based care* model adopts the WHO Package of Essential Non-communicable Disease Interventions (WHO PEN).[8] In this model, the Ministry of Health added the function of a health centre to *hospital-based care*. However, the coverage of health centres with the WHO PEN varies in each OD, which can be divided into low coverage (<50% of all health centres implement the WHO PEN; *health centre-based care (low)*) and high coverage (≥50% of all health centres implement the WHO PEN; *health centre-based care (high)*). The referral flow is slightly different between T2D and HTN. For T2D, this model identifies cases in the health centres through a screening test. If the patients are suspected of having T2D, health centres refer them to a diabetes clinic at a district referral hospital for confirmation of diagnosis. Once diagnosed, severe cases are treated in the hospital clinic, and stable or mild cases are followed up regularly at the health centres. For HTN, the health centres treat mild patients and refer the severe cases to the referral hospital. By 2021, *health centre-based care* was implemented in 137 of 1221 health centres.

⇒ The *community-based care* model or peer education network established and run by MoPoTsyo, a local non-governmental organisation.[8] In this model, peer educators (PEs) are added to hospital-based care. MoPoTsyo trained people with T2D and/or HTN to be PEs. These PEs play a role in screening and referring those suspected of having T2D and/or HTN to seek medical consultation and treatment at the referral hospitals that MoPoTsyo has partnered with. The PEs also provide counselling on lifestyle changes and support self-management to registered network members. By 2019, this *community-based care* model had been implemented in 20 of 102 ODs in 8 of 25 provinces in Cambodia.[8 32] It had 225 PEs to serve 40 000 people with T2D.[32]

⇒ The *coexisting care* model comprises a combination of the above three models. At the time of the study until 2021, only one OD (Daunkeo) had this model.

HTN, hypertension; T2D, type 2 diabetes.

---

be covered by a health post.[5] The health post provides similar services to a health centre, but it is smaller than a health centre.[5] Each OD usually has one district referral hospital with a few ODs having two district hospitals.[5] The district referral hospital receives self-referred patients or those referred by the health centres.

Alongside this public sector, a large *private healthcare sector* exists, which is more accessible than the public sector, and dominates outpatient curative care.[5] Since 1994, the Cambodian government started economic liberalisation, permitting staff to work outside their government's working hours and own healthcare facilities.[5] Since then, the private healthcare sector and dual practice system, meaning public healthcare workers also have private practices, have grown rapidly. In 2015, over 50% of the healthcare workforce in private healthcare facilities were

government personnel.[5] The private healthcare facilities range from cabinets, laboratories, pharmacies, clinics and polyclinics to hospitals.[5] Cabinets are the smallest facilities with less than two beds and mainly provide medical consultation services.[10 11] According to the MoH Progress Report in 2018, over 90% of private healthcare facilities were cabinets.[10 11] The second most frequent facilities were clinics (3.2%), providing medical specialties, laboratories, radiology services and pharmacies.[10] A clinic has between 10 and 20 beds.[10] In addition, buying medication in pharmacies or drugstores for self-treatment without a doctor's prescription is common in Cambodia, although not permitted by law.[12]

In terms of health expenditure, the public healthcare sector did not charge user fees until 1996.[5] In that year, the government introduced a user-fee scheme for the public sector with fees approved by the local community to increase healthcare quality at public healthcare facilities.[5 13] The revenue from the user-fee scheme could be used to incentivise staff and support ongoing operations. However, the user-fee posed challenges for the poor to access public healthcare. To address this, the MoH established the Health Equity Fund (HEF) in 2000, a pro-poor social health protection scheme.[14] The HEF is linked to the implementation of identification of the poor (known as 'IDPoor').[15] It is intended for the 'extremely poor' or 'poor' category, which is assessed and verified by the local authorities.[15] People with IDPoor are entitled to HEF support, meaning that they receive free healthcare services at public healthcare facilities and transportation expenditure reimbursement.[15] By 2019, the HEF covered approximately 3 million or about 20% of Cambodia's population.[14] Another scheme is the National Social Security Fund (NSSF), established in 2007.[16] The NSSF covers work and non-work-related illnesses and injuries for formally employed people.[16] Formal employers are mandated to pay for their staff's NSSF membership. The NSSF had enrolled over 1.7 million employees or about 11% of the population by 2019.[17]

However, it is important to note that several studies have indicated that the private sector still constitutes a significant source for out-of-pocket expenditure (OOPE).[18 19] Between 2009 and 2016, around 60% of health expenditure was OOPE while the rest was a combination of the government's and development partners' budgets. The OOPE per capita increased slightly from US$40.6 in 2009 to US$48.1 in 2016.[19] In 2016, 76.6% of the total OOPE was linked to private healthcare.[18 19]

## METHODS

### Data sources

This study is part of a larger cross-sectional household survey, with the primary aim of developing a care cascade for T2D and HTN.

### Settings

The study purposely selected five ODs. The selection was made to include different T2D and/or HTN care

models piloted in Cambodia: coexisting care, *community-based care*, *health-centre based care (high)*, *health-centre based care (low)*, and *hospital-based care* (box 1).

The five ODs in which the study took place are out of 103 ODs in the country and located in five different provinces. The map of ODs is presented in online supplemental annexure 1. These ODs have similar road infrastructure improvements, in which poor road conditions are no longer a barrier to accessing healthcare.

► *OD Daunkeo, Takeo province*. This OD had the 'coexisting care' model. At the time of the study, it was the only OD in which the three care models coexisted. The catchment area included Takeo town and a large rural area. Its non-communicable disease (NCD) clinic was established in 2002, and the peer educator network was initiated in 2007 and handed over to the MoH in 2015.[20] The WHO Package of Essential Non-communicable Disease Interventions (PEN) was implemented in 5 out of 14 health centres since 2015. The private services for people with T2D and/or HTN may also be easily accessible.

► *OD Kong Pisey, Kampong Speu province*. This OD had the 'community-based care' model. It has a strong MoPoTsyo network to provide T2D and HTN care to patients. Located about 54 km from the capital of Phnom Penh, this OD is semi-urban with a variety of private facilities.

► *OD Pearaing, Prey Veng province*. This OD had the 'health centre-based (high)' model, and was the OD with high coverage of the WHO PEN. Six out of nine health centres in this OD have been piloting the WHO PEN since 2015. Due to dual practice, the high coverage of the WHO PEN also facilitates accessible private services for people with T2D and/or HTN.

► *OD Sot Nikum, Siem Reap province*. This OD had the 'health centre-based (low)' model, and was the OD with low coverage of the WHO PEN (6/25 of the health centres started the WHO PEN in 2018). This OD has been historically and significantly influenced by the financial support of various development partners, and services for people with T2D and/or HTN have been well arranged at its NCD clinic.[21]

► *OD Samrong, Oddar Meanchey province*. This OD had a 'hospital-based care' model. It had an NCD clinic without the WHO PEN and peer educator network. A large part of the catchment area is a remote area bordering Thailand, approximately 470 km from the capital. Therefore, the private services for people with T2D and/or HTN may not be broadly accessible.

### Samples

The larger household survey recruited 5072 individuals aged 40 years or older to participate in the study using a multi-stage cluster sampling method. Initially, it purposively chose five ODs with different care models for T2D and HTN. Second, 44 villages per OD were randomly selected, regardless of the population size of each OD. The purpose of this equal probability selection was to

over-sample participants in ODs with a smaller population so that they would have an adequate sample for each care model. Third, 24 households in each village were selected by probability systematic sampling, and finally, one person aged 40 years or older per household was selected at random. To minimise the non-response rate, which can unintentionally exclude a certain group of the target population from the survey, the selected participants were called back or followed up three times when they were absent from their household. If the attempt failed, another household in the next row in the sampling list was selected. Then, the procedure described above was repeated. The equal probability selection at the village and household levels were used with the OD level's same purpose.

To correspond to our analytical objective, we used a subset of this sample: we only retained those who reported using healthcare services at least once in the 3 months preceding the survey (figure 1). A total of 2360/5072 participants met this criterion. The 2360-participant sample subset included four patient groups: 1331 people without T2D and HTN, 761 people with HTN alone, 109 people with T2D alone and 159 people with T2D plus HTN.

### Data collection
The data collection took place between July and October 2020. The data collection was conducted in three steps following the WHO's STEPS Survey approach: (1) interviews with a structured questionnaire, (2) anthropometric measurements and (3) biochemical measurements.[6] Since our study only focuses on healthcare usage and expenditure, we only used information from step 1—interviews with a structured questionnaire.

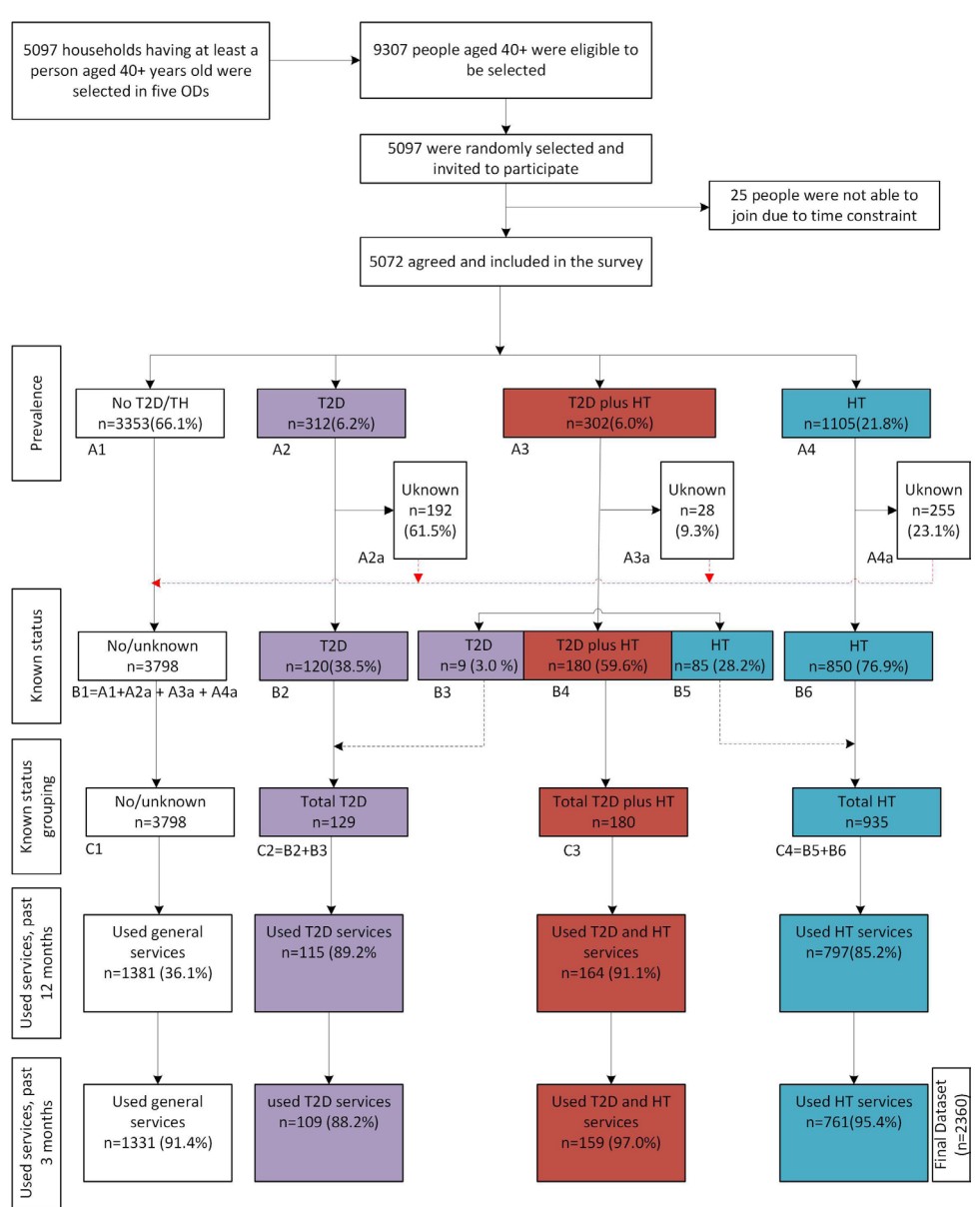

**Figure 1** Data flow from household selection to final dataset in this study, Cambodia, 2020. HT, hypertension; ODs, operational districts; T2D, type 2 diabetes.

The questionnaire was tablet-based and comprised 11 sections, including sociodemographic information, health status and quality of life, healthcare usage, social support, lifestyle or behaviour measures, physical activity, diabetes and hypertension knowledge, medication adherence, self-management support and decision-making power on food. However, we only used two sections in our analysis: sociodemographic information and healthcare usage.

The tablet-based questionnaire was installed using the Kobo Toolbox (https://kf.kobotoolbox.org), an open-source software with a free-of-charge server and online storage.[22]

## Measures

This study's primary variable of interest is the number of visits to public and private facilities.

By public healthcare facilities, we refer to government-run facilities that provide medical services, and include national hospitals, provincial referral hospitals, district referral hospitals, health centres and health posts. A health post is similar to a health centre, and only a few exist in remote areas. Therefore, we grouped them with health centres. Private healthcare services are non-government organisations that provide medical and non-medical services, and include private hospitals, private clinics, pharmacies, homes of trained health workers and visits of health workers to the patients' homes. Traditional healing/medicine and using healthcare services abroad have also been included in this category.

The secondary variable of interest was healthcare expenditure, the lump sum expenditure of medical consultation, treatment and medication. These data were obtained from the interview with the participants. They were asked about their use of health services in the 3 months preceding the survey (where they went, how often they went to a particular type of healthcare facility and how much they spent in each facility in those 3 months). We include the questionnaire in online supplemental annexure 2. The Cambodian currency (riels) was converted into US dollars (USD) at an exchange rate of 4000 riels per USD. The expenditure does not include other spendings such as on transport, food or guesthouses/hotels.

To better understand the profile of people using public or private healthcare facilities, we estimated associations between the use of public and private healthcare services and patient characteristics such as sex (male, female), age (40–49, 50–59, 60+ years old), educational level (none, primary, secondary or higher), social protection status (NSSF (yes, no), HEF (yes, no)), wealth quintile (poorest, poor, medium, rich, richest), type of care models (*hospital-based, health centre-based (high), health centre-based (low)* and *community-based*). The details on wealth quintile calculation (socioeconomic class) are described in online supplemental annexure 3.

## Analysis

### Healthcare usage

Taking the number of visits as a dependent variable, we report the healthcare visit rate to public and private facilities over the 3 months preceding the survey, then stratified by patient groups.

To identify the independent factors associated with healthcare usage (defined by the number of visits), we first used bivariate negative binomial regression to identify the potential factors in the five groups—overall and four patient groups—separately. Variables with a p value <0.25 in at least one of the four patient groups or overall group were included in the multiple negative binomial regression. The exposure variable (total healthcare visits of each participant) was incorporated into this model. Variables with a p value <0.05 were considered statistically significant in this final model. The negative binomial regression was chosen over Poisson regression because the number of visits was over-dispersed.

### Healthcare expenditure

We took healthcare expenditure in the 3 months preceding the survey as the dependent variable. Due to the limitation of our data, we focused more on assessing the factors associated with healthcare expenditure and did not explore the overall medical expenditure. We reported the overall arithmetic mean and then stratified the mean by patient groups. The expenditure was calculated separately for each patient group. Because arithmetic means can be easily affected by extremely high values, we removed the values above the 90th percentile, which we believe were too high in our sample.

Our analysis was carried out in three steps to separately identify the independent factors associated with healthcare expenditure in the four patient groups. First, a logarithmic transformation of the healthcare expenditure was performed as the data was skewed to the right. Second, in the bivariate analysis, we compared the geometric mean of healthcare expenditure by characteristics of the participants. This analysis identified the variables potentially associated with the healthcare expenditure at a p value <0.25. During this phase, the Student's t-test for binary explanatory variables and the one-way analysis of variance test for non-binary explanatory variables were used. Variables with a significant level at a p value <0.25 in any patient group were included in the multiple linear regression. Third, multiple linear regression was performed and the coefficient and 95% CI values were exponentiated to a risk ratio (RR) for better interpretation.

Data were analysed using Stata V.16.0 (Stata Corp LLC, College Station, Texas, USA), and R programing's GGPLOT2 package was used to produce the graphs.

## Patient and public involvement

No patient was involved in the development of the research question and outcome measures, study design and study participant recruitment. The findings are not disseminated to the study participants.

**Table 1** Demographic and socioeconomic characteristics of participant, 2020, Cambodia

| Variable | Overall (N=2360) | No T2D/HTN (N=1331) n (%) | HTN (N=761) n (%) | T2D (N=109) n (%) | T2D plus HTN (N=159) n (%) | P value |
|---|---|---|---|---|---|---|
| Sex of participant | | | | | | |
| Male | 689 (29.2) | 457 (34.3) | 179 (23.5) | 29 (26.6) | 24 (15.1) | <0.001 |
| Female | 1671 (70.8) | 874 (65.7) | 582 (76.5) | 80 (73.4) | 135 (84.9) | |
| Age in years | | | | | | |
| Range | 40–96 | 40–96 | 40–90 | 40–81 | 40–82 | |
| Mean (±SD) | 58.5 (± 10.4) | 56.0 (±10.3) | 62.4 (±10.0) | 57.6 (±8.4) | 61.7 (±8.2) | <0.001 |
| 40–49 | 497 (21.1) | 398 (29.9) | 75 (9.9) | 17 (15.6) | 7 (4.4) | <0.001 |
| 50–59 | 803 (34.0) | 464 (34.9) | 231 (30.4) | 46 (42.2) | 62 (39.0) | |
| 60 or older | 1060 (44.9) | 469 (35.2) | 455 (59.8) | 46 (42.2) | 90 (56.6) | |
| Educational level | | | | | | |
| No formal schooling | 757 (32.1) | 393 (29.5) | 283 (37.2) | 35 (32.1) | 46 (28.9) | <0.016 |
| Primary school | 1308 (55.4) | 755 (56.7) | 398 (52.3) | 61 (56.0) | 94 (59.1) | |
| Secondary school or higher | 295 (12.5) | 183 (13.7) | 80 (10.5) | 13 (11.9) | 19 (11.9) | |
| Having NSSF membership (yes) | 114 (4.8) | 62 (4.7) | 36 (4.7) | 6 (5.5) | 10 (6.3) | 0.806 |
| Having HEF membership (yes) | 434 (18.4) | 247 (18.6) | 143 (18.8) | 18 (16.5) | 26 (16.4) | 0.849 |
| Wealth quintile | | | | | | |
| Poorest | 441 (18.7) | 261 (19.6) | 140 (18.4) | 16 (14.7) | 24 (15.1) | 0.050 |
| Poor | 447 (18.9) | 263 (19.8) | 139 (18.3) | 18 (16.5) | 27 (17.0) | |
| Medium | 467 (19.8) | 262 (19.7) | 144 (18.9) | 27 (24.8) | 34 (21.4) | |
| Rich | 480 (20.3) | 244 (18.3) | 176 (23.1) | 16 (14.7) | 44 (27.7) | |
| Richest | 525 (22.2) | 301 (22.6) | 162 (21.3) | 32 (29.4) | 30 (18.9) | |
| Care model | | | | | | |
| Coexisting | 432 (18.3) | 248 (18.6) | 147 (19.3) | 20 (18.3) | 17 (10.7) | 0.015 |
| Community-based | 480 (20.3) | 276 (20.7) | 153 (20.1) | 18 (16.5) | 33 (20.8) | |
| Health centre-based (high) | 486 (20.6) | 257 (19.3) | 174 (22.9) | 27 (24.8) | 28 (17.6) | |
| Health centre-based (low) | 518 (22.0) | 292 (21.9) | 170 (22.3) | 18 (16.5) | 38 (23.9) | |
| Hospital-based | 444 (18.8) | 258 (19.4) | 117 (15.4) | 26 (23.9) | 43 (27.0) | |

Health centre-based (high) means the operational district (OD) with high coverage (six out of nine) of health centres with the WHO Package of Essential Non-communicable Disease Interventions (PEN); health centre-based (low) means the OD with low coverage (6 out of 25) of health centres with the WHO PEN.
HEF, Health Equity Fund; HTN, hypertension; NSSF, National Social Security Fund; T2D, type 2 diabetes.

## RESULTS
### Characteristics of participants
Our analysis included 2360 participants, including 1331 people without T2D or HTN, 761 people with HTN alone, 109 with T2D alone and 159 with T2D plus HTN (table 1). The other participants were excluded because they had not used healthcare services in the 3 months preceding the survey (N=2703) or had a missing response to the primary variable of interest (N=9).

As shown in table 1, females were more prevalent in all patient groups, especially in the T2D plus HTN group. The age range was between 40 and 96 years, with people with HTN and T2D plus HTN having a significantly higher average age than those without the two conditions. The majority of participants did not attend school or attended only primary school.

Regarding the social health protection scheme, a small proportion of participants in all groups had the NSSF membership (4.8% overall). A larger proportion of patients across all groups had the HEF membership (18.4% overall).

### Public and private healthcare usage
The 2360 individuals reported 6645 visits to the healthcare facilities in the 3 months preceding the survey, averaging 2.8 visits per person over 3 months.

Figure 2 presents the proportion of visits to public and private healthcare facilities. At the facility level, as

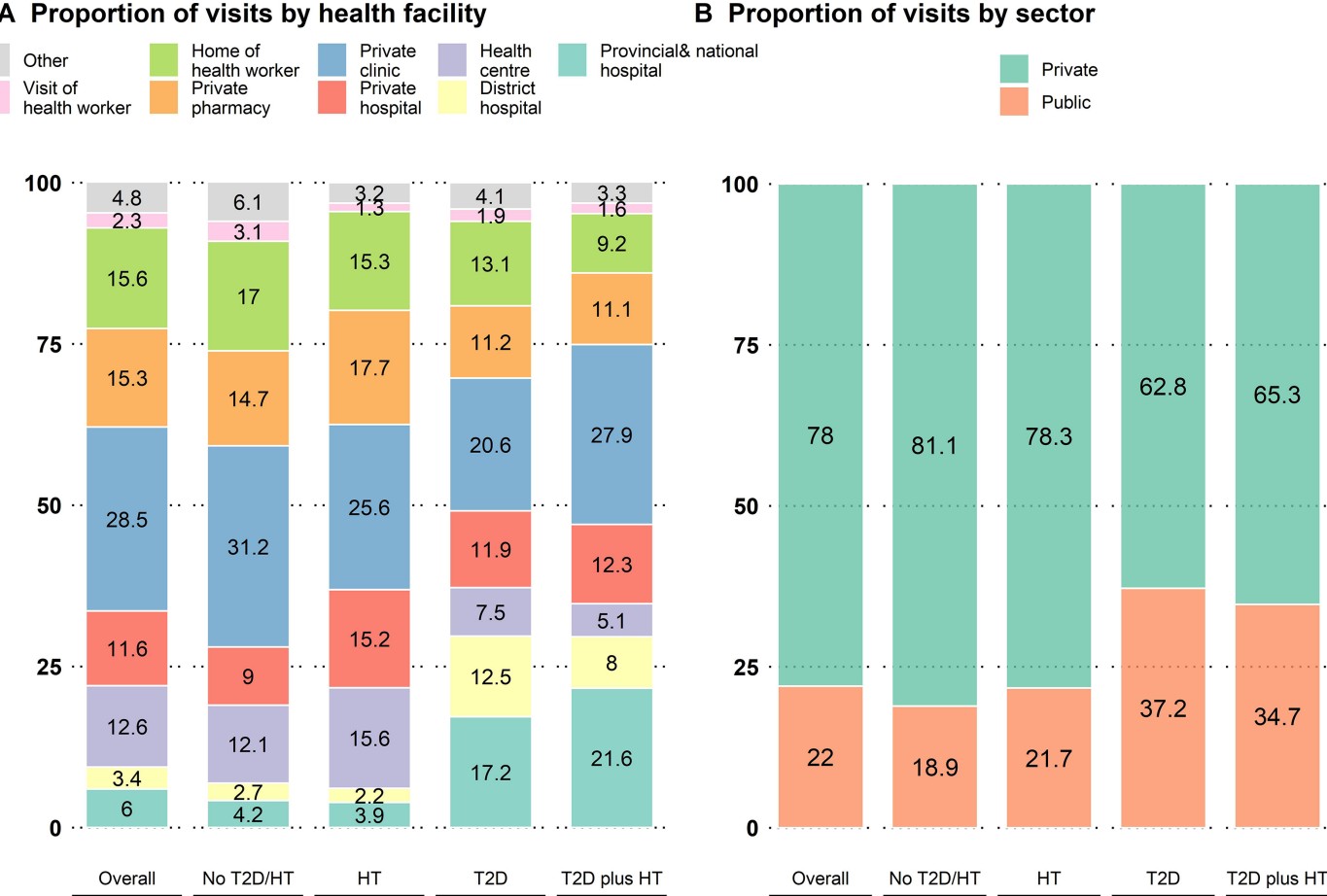

**A  Proportion of visits by health facility**

**B  Proportion of visits by sector**

Legend:
- Other
- Visit of health worker
- Home of health worker
- Private pharmacy
- Private clinic
- Private hospital
- Health centre
- District hospital
- Provincial & national hospital
- Private
- Public

**Figure 2**  Proportion of visits to public and private facilities in Cambodia, 2020. HT, hypertension; T2D, type 2 diabetes.

shown in figure 2A, the largest share was accounted for by private clinics (28.5%), followed by visits to the private homes of nurses or doctors (15.6%), private pharmacies (15.3%), health centres (12.6%), and private hospitals (11.6%).

The common public healthcare facilities used by participants with T2D and T2D plus HTN were provincial/national and district referral hospitals (figure 2A). Approximately 29.7% of visits from people with T2D and 29.6% from people with T2D plus HTN went to provincial/national and district hospitals (figure 2A). These proportions were higher than 6.9% for people without T2D or HTN and 6.1% for HTN only.

Overall, the private sector occupied about 78.0% of the total visits, and the public sector occupied 22.0% (figure 2B). All groups visited private healthcare facilities more frequently than public healthcare facilities (figure 2). However, the frequency of visiting public facilities was statistically higher in people with T2D and T2D plus HTN. As shown in figure 2B, 37.2% of visits from people with T2D and 34.7% of visits from people with T2D plus HTN were to public healthcare facilities, compared with 18.9% of visits from people without the two conditions and 21.7% of the visits from people with HTN (p value<0.001).

### Public healthcare usage by participant characteristics

Table 2 compares the public healthcare facility user rates defined as the proportion of public healthcare visits over total visits (public visits plus private visits). The user rates were disaggregated by participant characteristics. In this bivariate analysis, age, NSSF, HEF, wealth quintile and care model were significantly associated with public healthcare usage at a p value <0.25 in at least one patient group. Therefore, we included these variables in the multiple negative binomial regressions.

Table 3 presents the results of the multiple negative binomial regressions. Overall, the significant increase in public healthcare use was associated with having T2D and T2D plus HTN, living in the OD with *coexisting care*, and HEF membership. In people without T2D or HTN, HEF membership was significantly associated with public healthcare use: adjusted incidence rate ratio (AIRR) of 1.4 (95% CI 1.0 to 2.0), p value <0.05. We did not observe the same association in the other three groups.

In people with HTN, the poorest category was significantly associated with increasing public healthcare use with an AIRR of 2.1 (95% CI 1.1 to 4.0), p value=0.02, compared with those in the richest category. Nevertheless, the same association was not seen in other patient groups.

**Table 2** Proportions of visits to public facilities by participant characteristics, 2020, Cambodia

| Variable | Overall (N=6645) User rate (%) | No T2D/HTN (N=3467) User rate (%) | HTN (N=2345) User rate (%) | T2D (N=320) User rate (%) | T2D plus HTN (N=513) User rate (%) |
|---|---|---|---|---|---|
| **Sex** | | | | | |
| Male | 20.9 | 17.9 | 21.9 | 47.2 | 26.4 |
| Female | 23.7 | 20.7 | 23.2 | 33.3 | 37.6 |
| P value | 0.936 | 0.803 | 0.879 | 0.552 | 0.847 |
| **Age in years** | | | | | |
| 40–49 | 17.5 | 14.8 | 27.4 | 23.8 | 32.1 |
| 50–59 | 24.3 | 22.4 | 19.2 | 35 | 51.4 |
| 60+ | 24.0 | 21.1 | 24.2 | 43.5 | 26.3 |
| P value | 0.442 | **0.096** | 0.601 | 0.708 | 0.254 |
| **Educational level** | | | | | |
| No schooling | 21.9 | 20.1 | 19.3 | 37.5 | 39.6 |
| Primary | 22.9 | 18.8 | 23.2 | 45 | 34.6 |
| Secondary/higher | 25.5 | 22.6 | 33.1 | __ | 29.5 |
| P value | 0.998 | 0.981 | 0.606 | 0.92 | 0.932 |
| **Having NSSF membership** | | | | | |
| No | 22.2 | 19.9 | 21.2 | 37.5 | 33.3 |
| Yes | 35.9 | 17.1 | 52.4 | 31.6 | 66.7 |
| P value | 0.512 | 0.4505 | **0.165** | 0.808 | 0.343 |
| **Having HEF membership** | | | | | |
| No | 19.8 | 18 | 18.3 | 34.2 | 29.5 |
| Yes | 35.7 | 27.5 | 39.5 | 55.6 | 65.1 |
| P value | **<0.001** | **0.01** | **0.014** | 0.346 | **0.04** |
| **Household socioeconomic class** | | | | | |
| Poorest | 28.2 | 20.4 | 33.5 | 54.5 | 58 |
| Poor | 24.9 | 23.5 | 23.3 | 27.3 | 42.7 |
| Medium | 22.3 | 15.2 | 25.7 | 36.6 | 37.5 |
| Rich | 22.1 | 20.5 | 19.9 | 50 | 30 |
| Richest | 18.1 | 18.9 | 14.8 | 29.9 | 16.5 |
| P value | 0.029 | 0.419 | **0.083** | 0.966 | **0.218** |
| **Care model** | | | | | |
| Coexisting | 31.1 | 25.8 | 36.6 | 53.0 | 20.0 |
| Community-based | 19.5 | 18.0 | 13.6 | 56.8 | 39.2 |
| Health centre-based (high) | 14.7 | 13.2 | 14.8 | 19.0 | 23.8 |
| Health centre-based (low) | 21.4 | 21.5 | 16.6 | 46.7 | 32.7 |
| Hospital-based | 24.4 | 16.3 | 29.7 | 22.5 | 42.9 |
| P value | **0.002** | **0.007** | **0.018** | **0.172** | **0.162** |

The percentage of private healthcare is not presented in this table but can it be calculated by subtracting the percentage of the public healthcare from 100%. Health centre-based (high) means the operational district (OD) with high coverage (six out of nine) of health centres with the WHO Package of Essential Non-communicable Disease Interventions (PEN); health centre-based (low) means the OD with low coverage (6 out of 25) of health centres with the WHO PEN. 'N' denotes the total of visits. P values <0.25 are in bold, indicating a significant level at 0.25. Variables with p value <0.25 were included in multivariate analysis.
HEF, Health Equity Fund; HTN, hypertension; NSSF, National Social Security Fund; T2D, type 2 diabetes.

Regarding the care model, in people with T2D, the OD with *community-based care* (AIRR 3.7 (95% CI 1.2 to 11.3), p value=0.019) and the OD with low coverage of *health centre-based care* (AIRR 3.3 (95% CI 1.1 to 9.8), p value=0.036) were significantly higher in public healthcare use than in the OD with *hospital-based care*.

**Table 3** Factors associated with public healthcare use, 2020, Cambodia

| Disease group | Overall (N=2360) AIRR (95% CI) | No T2D/HTN (N=1331) AIRR (95% CI) | HTN (N=161) AIRR (95% CI) | T2D (N=109) AIRR (95% CI) | T2D plus HTN (N=759) AIRR (95% CI) |
|---|---|---|---|---|---|
| No T2D/HTN | Ref. | – | – | –_ | – |
| HTN | 1.0 (0.8 to 1.2) | – | – | – | – |
| T2D | **1.9 (1.3 to 2.9)\*\*** | – | – | – | – |
| T2D plus HTN | **1.9 (1.3 to 2.7)\*\*\*** | – | – | – | – |
| Age in year | | | | | |
| 40–49 | Ref. | Ref. | Ref. | Ref. | |
| 50–59 | 1.2 (0.9 to 1.5) | 1.4 (1.0 to 1.9) | 0.6 (0.3 to 1.2) | 1.5 (0.5 to 4.5) | 1.2 (0.3 to 4.2) |
| 60+ | 1.1 (0.8 to 1.4) | 1.2 (0.9 to 1.7) | 0.7 (0.4 to 1.3) | 1.5 (0.5 to 4.5) | 0.6 (0.2 to 2.2) |
| Having NSSF membership | | | | | |
| No | Ref. | Ref. | Ref. | Ref. | |
| Yes | 1.4 (0.9 to 2.1) | 1.0 (0.5 to 1.8) | 2.0 (0.9 to 4.6) | 2.3 (0.5 to 10.1) | 1.9 (0.7 to 4.8) |
| Having HEF membership | | | | | |
| No | | Ref. | Ref. | Ref. | |
| Yes | **1.4 (1.1 to 1.8)\*** | **1.4 (1.0 to 2.0)\*** | 1.4 (0.9 to 2.3) | 2.1 (0.8 to 5.1) | **1.9 (1.0 to 3.7)** |
| Household socioeconomic class | | | | | |
| Poorest | **1.4 (1.0 to 2.0)\*** | 1.0 (0.7 to 1.6) | **2.1 (1.1 to 4.0)\*** | 1.2 (0.4 to 3.5) | 2.6 (0.9 to 7.1) |
| Poor | 1.2 (0.9 to 1.7) | 1.1 (0.7 to 1.7) | 1.2 (0.7 to 2.3) | 0.7 (0.2 to 2.1) | 2.6 (1.0 to 7.3) |
| Medium | 1.1 (0.8 to 1.5) | 0.9 (0.6 to 1.4) | 1.2 (0.7 to 2.3) | 0.9 (0.4 to 2.2) | 3.0 (1.2 to 7.7) |
| Rich | 1.1 (0.8 to 1.4) | 1.1 (0.7 to 1.7) | 1.0 (0.6 to 1.8) | 1.2 (0.4 to 3.5) | 1.7 (0.7 to 4.1) |
| Richest | Ref. | Ref. | Ref. | Ref. | Ref. |
| Care model | | | | | |
| Coexisting | **1.4 (1.0 to 1.9)\*** | 1.5 (1.0 to 2.3) | 1.3 (0.7 to 2.3) | 2.5 (0.8 to 7.6) | **4.0 (1.2 to 12.9)\*** |
| Community-based | 0.9 (0.7 to 1.2) | 1.0 (0.6 to 1.5) | 0.5 (0.3 to 1.0) | **3.7 (1.2 to 11.3)\*** | 1.7 (0.5 to 6.1) |
| Health centre-based (high) | 0.8 (0.6 to 1.1) | 0.8 (0.5 to 1.3) | 0.7 (0.4 to 1.3) | 1.3 (0.4 to 3.9) | 2.7 (0.9 to 8.8) |
| Health centre-based (low) | 1.0 (0.8 to 1.4) | 1.4 (1.0 to 2.1) | 0.6 (0.3 to 1.0) | **3.3 (1.1 to 9.8)\*** | 3.0 (1.0 to 9.1) |
| Hospital-based | Ref. | Ref. | Ref. | Ref. | Ref. |

Health centre-based (high) means the OD with high coverage (six out of nine) of health centres with the WHO Package of Essential Non-communicable Disease Interventions (PEN); health centre-based (low) means the OD with low coverage (6 out of 25) of health centres with the WHO PEN.
\*p<0.05, \*\*p<0.01, \*\*\*p<0.001.
AIRR, adjusted incidence rate ratio; HEF, Health Equity Fund; HTN, hypertension; NSSF, National Social Security Fund; OD, operational district; Ref., reference group; T2D, type 2 diabetes.

In people with T2D plus HTN, the OD with *coexisting care* was associated with higher public healthcare use (AIRR 4.0 (95% CI 1.2 to 12.9), p value=0.020).

### Healthcare expenditure

#### Medical cost per year, overall and by facility

Overall, those who used healthcare spent an average of US$25.3 (95% CI 22.9 to 27.6) for all healthcare services in the 3 months preceding the survey (figure 3).

When comparing patient groups, people with T2D plus HTN had the highest healthcare expenditure with an average of US$43.6 (95% CI 29.7 to 57.2), followed by people with T2D with an average of US$34.0 (95% CI 25.5 to 42.6). These expenditures were statistically higher than the average of US$17.1 (95% CI 13.1 to 21.1) in people with HTN and the average of US$26.9 (95% CI 23.9 to 29.9) in people without the two conditions with a p value <0.001.

Online supplemental annexure 4 table S1 shows the arithmetic mean of healthcare expenditure. The arithmetic mean is the mean before the data log-transformation. Since our model's RR in table 4 is the geometric mean (after log-transformation) ratio, we presented the geometric mean in online supplemental annexure 4 table S2. In the bivariate analysis, sex, age, NSSF, HEF, wealth quintile, sector (public vs private) and care model were statistically associated with healthcare expenditure in one or more patient groups with a p value <0.25. These variables were included in the multiple linear regression.

Table 4 presents results from the multiple linear regression analyses. Overall, having T2D or T2D plus HTN, being female, having reported using both private and public healthcare, and living in the OD with *community-based care* was significantly associated with increased healthcare

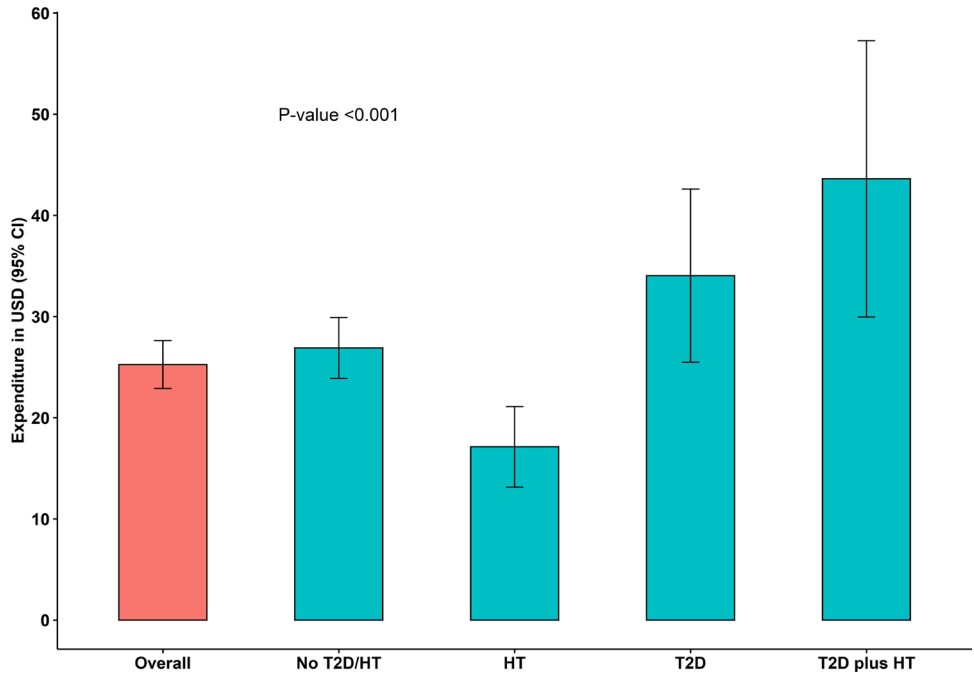

**Figure 3** Healthcare expenditure by patient groups in the 3 months preceding the survey in 2020, Cambodia. HT, hypertension; T2D, type 2 diabetes.

expenditure. In contrast, holding HEF membership and using public healthcare was significantly associated with healthcare expenditure reduction.

At the group level, in people without the two conditions, HEF membership was significantly associated with a reduction in healthcare expenditure with an adjusted RR (ARR) of 0.7 (95% CI 0.5 to 0.8), p value <0.001. The same association was seen in people with HTN (ARR of 0.8 (95% CI 0.6 to 1.0), p value <0.01), and in T2D plus HTN (ARR of 0.3 (95% CI 0.2 to 0.6), p value <0.001). However, the association was not observed in people with T2D.

In people without the two conditions, using public healthcare was significantly associated with a reduction in the expenditure (ARR of 0.3 (95% CI 0.2 to 0.3), p value <0.001). The association was also found in people with HTN (ARR 0.4 (95% CI 0.3 to 0.5), p value <0.001).

People with T2D plus HTN who resided in the OD with *community-based care* were significantly associated with a higher expenditure with an ARR of 2.0 (95% CI 1.1 to 3.8), p value <0.01 than those with *hospital-based care*.

## DISCUSSION

The results show that the use of the public healthcare system remains low for all groups in our study, with about one in every five healthcare visits taking place in the public sector overall. People with chronic conditions, HEF membership, living in the OD with *community-based care* contributed to public healthcare uptake. The healthcare expenditure was significantly reduced when patients used public healthcare services, regardless of HEF membership. However, the reduction in spending was more noticeable in people with HEF membership. In

contrast, expenditure was higher among patients living in the OD with *community-based care.*

People in Cambodia predominantly used healthcare in private facilities for outpatient curative care.[19] Our study showed that this is also the case for people with chronic conditions such as T2D and HTN, although this group had a slightly higher rate of using public healthcare services. This result is congruent with earlier findings that approximately 61% of T2D and/or HTN patients received their initial diagnosis in private settings.[7] A qualitative study in Cambodia suggested that people with T2D did not prefer diabetes services at public facilities because they were less accessible due to geographical factors or distance and limited medication supply.[23] Comparing our findings to other low-income and middle-income countries is challenging due to differences in health system organisation, government investment in health and most studies focusing on general services rather than T2D and/or HTN services. Nevertheless, our findings are comparable to those from India, Nigeria and Nepal, where government accounts for a very small share (<30%) of national health expenditure as well.[24] In India, 75% of outpatient visits were occupied by the private sector, similar to Nigeria (82%) and Nepal (65%).[24] Cambodia and these three countries shared similar characteristics as the majority of the population relies on low-cost, and low-quality private services. Our findings clearly suggest that healthcare quality and access to public healthcare services is still below the expectations of people and the private providers have a role in filling the gaps. To move forward in the direction of universal health coverage, meaning that people can access the health services they need without financial hardship, Cambodia should focus

**Table 4** Factors associated with reducing or increasing healthcare expenditure in 2020, Cambodia

| Variable | Overall (N=2142) ARR (95% CI) | No T2D/HTN (N=1187) ARR (95% CI) | HTN (N=726) ARR (95% CI) | T2D (N=98) ARR (95% CI) | T2D plus HTN (N=139) ARR (95% CI) |
|---|---|---|---|---|---|
| Disease group | | | | | |
| No T2D/HTN | Ref. | – | – | – | – |
| HTN | 0.9 (0.7 to 1.0) | – | – | – | – |
| T2D | **2.1 (1.6 to 2.7)*** | – | – | – | – |
| T2D plus HTN | **1.9 (1.5 to 2.4)*** | – | – | – | – |
| Sex | | | | | |
| Male | | Ref. | Ref. | Ref. | Ref. |
| Female | **1.2 (1.1 to 1.4)** | **1.4 (1.2 to 1.7)*** | 1.0 (0.8 to 1.2) | 0.7 (0.4 to 1.5) | 1.5 (0.8 to 2.9) |
| Age in years | | | | | |
| 40–49 | Ref. | Ref. | Ref. | Ref. | |
| 50–59 | 1.1 (0.9 to 1.3) | 1.0 (0.8 to 1.2) | 1.4 (1.0 to 2.0) | 1.1 (0.5 to 2.4) | 1.4 (0.5 to 3.7) |
| 60+ | 1.0 (0.9 to 1.2) | 1.1 (0.8 to 1.3) | 1.3 (1.0 to 1.9) | 0.9 (0.4 to 1.9) | 1.0 (0.4 to 2.7) |
| Educational level | | | | | |
| No schooling | Ref. | Ref. | Ref. | Ref. | Ref. |
| Primary | 1.0 (0.9 to 1.1) | 1.1 (0.9 to 1.4) | 0.8 (0.7 to 1.0) | 0.9 (0.5 to 1.7) | 1.1 (0.7 to 1.8) |
| Secondary/higher | 1.1 (0.8 to 1.3) | 1.2 (0.9 to 1.6) | 0.9 (0.6 to 1.3) | – | 1.6 (0.7 to 3.7) |
| Having NSSF membership | | | | | |
| No | Ref. | Ref. | Ref. | Ref. | Ref. |
| Yes | 0.9 (0.6 to 1.2) | 0.9 (0.6 to 1.4) | 0.9 (0.6 to 1.4) | 0.6 (0.2 to 2.0) | **0.4 (0.2 to 0.9)*** |
| Having HEF membership | | | | | |
| No | Ref. | Ref. | Ref. | Ref. | Ref. |
| Yes | **0.7 (0.6 to 0.8)*** | **0.7 (0.5 to 0.8)*** | **0.8 (0.6 to 1.0)*** | 0.7 (0.3 to 1.6) | **0.3 (0.2 to 0.6)*** |
| Household wealth quintile | | | | | |
| Poorest | 1.1 (0.9 to 1.4) | 1.3 (0.9 to 1.7) | 1.1 (0.8 to 1.5) | 0.5 (0.2 to 1.2) | 1.1 (0.5 to 2.2) |
| Poor | 1.1 (0.9 to 1.3) | 1.1 (0.8 to 1.4) | 1.2 (0.9 to 1.7) | 0.8 (0.3 to 1.8) | 0.6 (0.3 to 1.3) |
| Medium | 1.1 (0.9 to 1.3) | 1.1 (0.8 to 1.4) | 1.2 (0.9 to 1.6) | 0.8 (0.4 to 1.6) | 0.9 (0.4 to 1.7) |
| Rich | 1.2 (1.0 to 1.4) | 1.3 (1.0 to 1.7) | 1.2 (0.9 to 1.6) | 0.7 (0.3 to 1.7) | 0.9 (0.5 to 1.7) |
| Richest | Ref. | Ref. | Ref. | Ref. | Ref. |
| Healthcare sector | | | | | |
| Private | Ref. | Ref. | Ref. | Ref. | Ref. |
| Public | **0.3 (0.3 to 0.4)*** | **0.3 (0.2 to 0.3)*** | **0.4 (0.3 to 0.5)*** | 0.7 (0.4 to 1.2) | 0.8 (0.5 to 1.3) |
| Both | **1.5 (1.1 to 1.9)*** | **1.6 (1.1 to 2.4)** | 0.9 (0.4 to 1.7) | – | 1.2 (0.3 to 5.4) |
| OD with different care | | | | | |
| Coexisting | 1.0 (0.9 to 1.3) | 1.0 (0.8 to 1.3) | 1.0 (0.7 to 1.4) | 1.0 (0.4 to 2.6) | 1.7 (0.8 to 3.9) |
| Community-based | **1.4 (1.1 to 1.6)** | 1.2 (0.9 to 1.6) | 1.3 (1.0 to 1.9) | 1.3 (0.6 to 3.2) | **2.0 (1.1 to 3.8)** |
| Health centre-based (high) | 1.0 (0.8 to 1.2) | 1.0 (0.7 to 1.3) | 1.1 (0.8 to 1.5) | 0.9 (0.4 to 1.9) | 1.3 (0.7 to 2.6) |
| Health centre-based (low) | 0.8 (0.7 to 1.0) | 0.8 (0.6 to 1.0) | 0.8 (0.6 to 1.1) | 1.3 (0.5 to 3.0) | 1.5 (0.8 to 2.6) |
| Hospital-based | Ref. | Ref. | Ref. | Ref. | Ref. |

*p<0.05, **p<0.01, ***p<0.001.
Health centre-based (high) means the operational district (OD) with high coverage (six out of nine) of health centres with the WHO Package of Essential Non-communicable Disease Interventions (PEN); health centre-based (low) means the OD with low coverage (6 out of 25) of health centres with the WHO PEN; p values <0.05, <0.01, <0.001 are in bold, indicating the variables are significantly associated with expenditure.
ARR, adjusted risk ratio; HEF, Health Equity Fund; HTN, hypertension; NSSF, National Social Security Fund; OD, operational district; Ref., reference group; T2D, type 2 diabetes.

on expanding quality service coverage for people with T2D and/or HTN at public healthcare facilities across the country. Expanding quality services at public healthcare facilities may be the best suited approach to the Cambodian context, where dual practice is strong and regulation weak. The literature suggests that the public and private providers are not mutually exclusive and they shape each other's characteristics or sometimes so-called

competition for health benefits.[24–27] If public healthcare providers can provide quality services at affordable prices to the poor or those from low-income households, visits to private healthcare providers, who offer inferior services at higher prices, will decrease.[24–27] The private healthcare providers will change their service provision to target the rich.[24–27] Previous studies revealed that HEF membership contributed to the health service uptake at public facilities and reduced healthcare expenditure in general users.[14 28] Our findings extended the understanding that HEF membership has also increased public healthcare use and substantially reduced healthcare spending among people with T2D and/or HTN. Since HEF benefits are only available in public healthcare facilities, it is not surprising that it also contributes to increasing service uptake in public facilities. The HEF is an important pillar of the Cambodian government's social security system and our findings suggest that HEF membership should be expanded to cover among people with chronic conditions. The Cambodian government recognised that the current social protection system has not yet covered those so-called 'missing middle' between the poor, who are covered by the HEF, and those in formal employment, who are covered by the NSSF. Therefore, a new social health protection scheme targeting those in the informal economy and senior population without pensions, which accounts for 90% of people aged 60 years or older in Cambodia, should be created. This social protection scheme must go alongside with improving service coverage and quality. The success of such a model has been demonstrated by Thailand, a neighbouring country of Cambodia.[29 30] Thailand focused on improving public healthcare services and introduced three public health insurance schemes. One of them was the Universal Coverage Scheme, which covered 75% of the Thai population.[29] Such a model might be too ambitious for Cambodia, since Thailand is more economically developed than Cambodia. However, this is still a model that Cambodia should be aiming for, so that quality health services for people with T2D and/or HTN will be more accessible.

*Community-based care* contributed to the higher public service uptake among T2D and/or HTN, but it also contributed to the higher expenditure for the users. In ODs with this model, peer educators (PEs) refer patients to the public referral hospitals, so it is not surprising that the public service uptake is slightly higher than other ODs.[31] However, it is somewhat surprising that people with T2D and/or HTN in the OD with *community-based care* spent more on their health services. It is unclear what the influencing factors are because a large proportion of service users (80%) used private services in this OD. Although, this may be partially explained by higher unit costs spent by the supply side in *community-based care* to operate their services, so the patients are charged a higher fee than other models. Our team had conducted a costing study in 2020 to examine the costs to operate services by different care models. The study found that the annual unit costs were higher for T2D and HTN patients in the *community-based care* than the *hospital-based care* (US$101 vs US$77 for a T2D patient and US$83 vs US$55 for an HTN patient). The higher unit costs in the *community-based care* were driven by adding PE components and field activities to the model while drugs and consultation fee are not subsidised. The investment in *community-based care* leads to better treatment outcomes, but it is not explored in our study. A previous study provided limited information that a significant proportion of patients in the *community-based care* network had achieved fasting blood glucose goals of 126 mg/dL, from 10% to 45%, and blood pressure goals of 140/90 mm Hg, from 58% to 67%, after a 12-month follow-up.[20] This study, however, did not have a control group (patients outside the network). From this, we can learn two things. First, the adapting and scaling up of PEs should be done with a careful budget plan as PEs incur operational costs. Second, a study investigating the treatment outcomes and cost-effectiveness between different care models should be conducted in order to inform decision-making. We, therefore, cannot make a recommendation from this limited finding.

There are several strengths in our study. First, our study is among the few to examine healthcare usage and expenditure both among people with T2D and/or HTN and people without the two conditions in Cambodia. It furthermore covers both the public and private sectors. This broad scope renders the results useful to inform T2D and HTN interventions in Cambodia. Second, we covered a wide range of ODs which are geographically diverse and comprising different care models, which means that our participants are heterogeneous. The sampling design—randomising villages, households and household members—is robust within its scope, targeting the population in rural or semi-rural settings in Cambodia. Third, the data collection was robust and ensured a reliable data set.

Our study also had its limitations. First, it may not represent the national level as most of the study sites (villages) we selected were rural or semi-rural, which may lead to overestimating the healthcare usage in public facilities. Second, the ODs were purposively selected with oversampling the OD with interventions, increasing the service uptake in public facilities. This may lead to overestimating the public healthcare use in our study. Third, we only calculated the healthcare expenditure for those who used the service in the 3 months preceding the survey, which cannot be generalised outside this period. However, it is unlikely to be significant because we focused more on factors associated with increasing or reducing healthcare expenditure. Fourth, the sample size for people with T2D only and people with T2D plus HTN may be relatively small. Therefore, variables that were not significantly associated with the dependent variables in these groups in our study may be due to the insufficient sample size.

## CONCLUSION

Healthcare usage at public healthcare facilities is relatively low for all groups; however, it is higher in people with chronic conditions. HEF membership and *community-based care* contributed the higher public healthcare usage in people with chronic conditions. Using public healthcare services, regardless of HEF status, reduced the healthcare expenditure. However, the reduction in spending was more noticeable in people with HEF membership. To protect people with T2D and/or HTN from financial risk and move in the direction of universal health coverage, the public healthcare system should further improve care quality, and expand social health protection. Future research should link healthcare use and expenditure across different healthcare models to actual treatment outcomes to denote areas for further investment.

**Author affiliations**
[1]Technical Office, National Institute of Public Health, Phnom Penh, Cambodia
[2]School of Public Health, National Institute of Public Health, Phonm Penh, Cambodia
[3]Department of Public Health, Prince Leopold Institute of Tropical Medicine, Antwerpen, Belgium
[4]Department of Family Medicine and Population Health, University of Antwerp, Antwerp, Belgium
[5]Centre for Population, Family & Health, University of Antwerp, Antwerpen, Antwerp
[6]Department of Social Sciences, University of Antwerp, Antwerpen, Belgium
[7]Department of Public Health, Institute of Tropical Medicine, Antwerpen, Belgium
[8]Gerontology, Vrije Universiteit Brussel, Brussels, Belgium
[9]Management team, National Institute of Public Health, Phonm Penh, Cambodia

**Correction notice** This article has been corrected since it was published online. Author Vannarath Te is also affiliated to affiliation 4.

**Acknowledgements** We would like to thank all the participants in the study and all data collectors for carefully following the instructions.

**Contributors** SChhim, IP, VT, EW designed the study. SChhim analysed data. SChhim, VT, VB, JvO, SChham, SL, SY, WvD, EW, IP wrote the manuscript.IP is the guarantor of this study who accepts full responsibility for this work, had access to the data, and controlled the decision to publish.

**Funding** This work was funded by the Belgian Government through the VLIR-UOS project (no award/grant number), the Fourth Funding Agreement (FA4) Programme (2017–2021) between the Belgian Directorate of Development Cooperation and the Institute of Tropical Medicine, Antwerp (no award/grant number), and the Horizon2020 Framework Programme of the European Union, grant no 825432.

**Map disclaimer** The inclusion of any map (including the depiction of any boundaries therein), or of any geographic or locational reference, does not imply the expression of any opinion whatsoever on the part of BMJ concerning the legal status of any country, territory, jurisdiction or area or of its authorities. Any such expression remains solely that of the relevant source and is not endorsed by BMJ. Maps are provided without any warranty of any kind, either express or implied.

**Competing interests** None declared.

**Patient and public involvement** Patients and/or the public were not involved in the design, or conduct, or reporting, or dissemination plans of this research.

**Patient consent for publication** Consent obtained directly from patient(s)

**Ethics approval** This study involves human participants and the protocol was approved by the National Ethics Committee for Human Research (NECHR) on 29 April 2019 (No. 105 NECHR) and by the Institutional Review Board of Institute of Tropical Medicine (Antwerp) on 25 October 2019 (No. 1323/19). Participants gave informed consent to participate in the study before taking part.

**Provenance and peer review** Not commissioned; externally peer reviewed.

**Data availability statement** Data are available upon reasonable request. Data are available on reasonable request. Data are available on reasonable request to IP ( ipor@niph.org.kh).

**Open access** This is an open access article distributed in accordance with the Creative Commons Attribution 4.0 Unported (CC BY 4.0) license, which permits others to copy, redistribute, remix, transform and build upon this work for any purpose, provided the original work is properly cited, a link to the licence is given, and indication of whether changes were made. See: https://creativecommons.org/licenses/by/4.0/.

**ORCID iDs**
Srean Chhim http://orcid.org/0000-0003-1558-1875
Vannarath Te http://orcid.org/0000-0002-2059-3283
Veerle Buffel http://orcid.org/0000-0001-6602-9525
Edwin Wouters http://orcid.org/0000-0003-2268-3829

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
