## [Reviewer comments · BMJ Open]

ARTICLE DETAILS

TITLE (PROVISIONAL)	Healthcare utilization and expenditure among people with type 2 diabetes and/or hypertension in Cambodia: results from a cross-sectional survey
AUTHORS	Chhim, Srean; Te, Vannarath; Buffel, Veerle; van Olmen, Josefien; Chham, Savina; Long, Sereyraksmeay; Yem, Sokunthea; Van Damme, Wim; Wouters, Edwin; Por, Ir

VERSION 1 – REVIEW

REVIEWER	Ramallo-Fariña , Yolanda Fundación Canaria de investigación Sanitaria, FUNCANIS, Servicio de Evaluación del Servicio Canario de la Salud
REVIEW RETURNED	18-Mar-2022

GENERAL COMMENTS	In general This is a good article that describes in detail the reality of healthcare in Cambodia for patients with diabetes and/or hypertension. In addition, the article show the factors associated with public healthcare use. Methods: The method used for the objective is appropriate in the context of the country (Cambodia), since it would have been desirable to have clinical information of the patients obtained through registries, but I understand that this is not feasible. The statistical methods used are appropriate. Results: In results should be related the expenditure in health care with the patient's health status. The health status could be defined by the survey question Q15 (VAS). Patient satisfaction (mesasured in the survey) with the service received should be related the expenditure in health care too. Discussion in the discussion should be related through the data provided by the literatura (if the exist) the clinical health outcomes of patients in Cambodia (Hba1c, blood pressure, BMI, ..) according to the different models of care.
---

REVIEWER	Nigenda, Gustavo National Autonomous University of Mexico, National School of Nursing and Obstetrics
-----------------	---

GENERAL COMMENTS

Healthcare utilization and expenditure among people with type 2 diabetes and/or hypertension in Cambodia: results from a cross-sectional survey

Summary:

The study provides an assessment of health services utilization and expenditure of people with Type 2 Diabetes and hypertension in five operational districts in Cambodia.

General opinion:

The results present an interesting picture on how population confronts their need for care against chronic diseases according to the options provided by a health care system which has applied over time a set of regressive policies that do not provide real protection to the population. However, the discussion fails to provide clues to modify structures and mechanisms to increase protection mainly proposing that a potential step forward is to combine the capacities of public and private sectors without explaining how this should/could be done.

Specific considerations:

- a) The starting section (context) of “methods and context” should be removed from this section and transferred to an “introduction and background” section as it provides information that is relevant to the whole paper (particularly the discussion section) and not only to the methods section.
- b) Regarding the care models: In the hospital-based care model the populations refer themselves to the units or are transferred by an intermediary unit?
- c) The issue of a government imposing user fees to poor populations and later on creating a program (HEF) to protect population against the pernicious effects of the original policy should invite authors to reflect on what to do with this entangle in the future. Would you suggest to abolish user fees as many countries did and to create a comprehensive fund that could cover all those who are not affiliated to NFFS?
- d) In a period where GDP growth rates of most countries are far below, growing at 7% annually is a great opportunity to recommend to the government that it should invest in health and education. Particularly in health to create a real public system. Authors should provide an opinion about this in the discussion.
- e) It is striking to see how an organizational improvement in the model of care (community based care model) would mean transferring added costs to the users. The public systems is acting as a private enterprise. Could authors provide and opinion about the risks of expanding new models that are more costly to the users.
- f) Authors make no comparisons whatsoever with similar studies in other countries. Discussing with other countries realities may open up to new ways of interpreting results of the Cambodian reality.
- g) Suggesting that a public private mix could be a next step forward without suggesting which could be specific characteristics of this mix leaves the readers with more questions than answers about how this could be a protective option for the population. International experiences on public private mix have shown a broad variety of results depending on how this is planned and implemented. Please comment about.

Comments	Responses
Reviewer: 1 Dr. Yolanda Ramallo-Fariña , Fundación Canaria de investigación Sanitaria, FUNCANIS Comments to the Author:	
In general This is a good article that describes in detail the reality of healthcare in Cambodia for patients with diabetes and/or hypertension. In addition, the article show the factors associated with public healthcare use.	Thank you for your encouraging comments.
Methods: The method used for the objective is appropriate in the context of the country (Cambodia), since it would have been desirable to have clinical information of the patients obtained through registries, but I understand that this is not feasible. The statistical methods used are appropriate.	Thanks for agreeing that the methodology used is appropriate.
Results: In results should be related the expenditure in health care with the patient's health status. The health status could be defined by the survey question Q15 (VAS). Patient satisfaction (mesasured in the survey) with the service received should be related the expenditure in health care too.	Thank you for this valuable suggestion. The authors agree that relating the expenditure to the patient's health status and patient's satisfaction could generate potentially interesting insights into the relationship between the expenditure and the quality of the care provided. However, this falls outside of the scope of our article which explicitly assesses the utilization of public and private healthcare, related healthcare expenditure, and associated factors for people with type 2 diabetes (T2D) and/or hypertension (HTN) and for people without those conditions in Cambodia. This is already a sufficiently broad focus, resulting in a manuscript of considerable length. We do however thank the reviewer for these suggestions, and the research team will comprehensively address these topics in future publications.

Discussion in the discussion should be related through the data provided by the literatura (if the exist) the clinical health outcomes of patients in Cambodia (Hba1c, blood pressure, BMI, ..) according to the different models of care.	Thanks for this valuable suggestion. We have extensively searched for relevant clinical health outcomes for the Cambodian population to relate our results to. However, we believe that such representative data simply does not exist. We do however agree that this is an important point. We, therefore, include a recommendation for future studies on the relationship between health expenditure and clinical health outcomes in the country. In the manuscript (clean version), please see lines 513-515 and lines 546-547.

Reviewer: 2 Dr. Gustavo Nigenda, National Autonomous University of Mexico Comments to the Author	
Summary: The study provides an assessment of health services utilization and expenditure of people with Type 2 Diabetes and hypertension in five operational districts in Cambodia.	
General opinion: The results present an interesting picture on how population confronts their need for care against chronic diseases according to the options provided by a health care system which has applied over time a set of regressive policies that do not provide real protection to the population. However, the discussion fails to provide clues to modify structures and mechanisms to increase protection mainly proposing that a potential step forward is to combine the capacities of public and private sectors without explaining how this should/could be done.	Thanks for bringing up important points that we missed. We have tried to improve our recommendations based on the lessons learned from other settings. We revised the recommendation related to the social protection scheme, community-based care, and the public-private mix.
Specific considerations: a) The starting section (context) of “methods and context” should be removed from this section and transferred to an “introduction and background” section as it provides information	Thank you for the suggestion. We removed the context from “Methods and context.” We, however, wish to keep the word “Context” as the

that is relevant to the whole paper (particularly the discussion section) and not only to the methods section.	sub-section of the introduction. We believe “Context” describes the sub-section better than “Background.” In the manuscript (clean version), please see lines 102-151.
b) Regarding the care models: In the hospital-based care model the populations refer themselves to the units or are transferred by an intermediary unit?	This is a valid question. The answer is that the patients refer themselves to the units (and they are thus not transferred by an intermediary unit). We have added this information to the manuscript text. Please see Box 1 starting from line 92.
c) The issue of a government imposing user fees to poor populations and later on creating a program (HEF) to protect population against the pernicious effects of the original policy should invite authors to reflect on what to do with this entangle in the future. Would you suggest to abolish user fees as many countries did and to create a comprehensive fund that could cover all those who are not affiliated to NFFS?	Thanks for bringing a key point to our paper. The HEF is an important pillar of the government's social security system. The abolishing of user fees may not be the best choice for the Cambodian government at this stage because Cambodian economics is still weak. The service cannot be free of charge for everyone, but those financially vulnerable should be protected. The Cambodian government tried to improve and expand its social protection scheme guided by the National Social Protection Policy Framework 2016-2025. Even though they have resource challenges, they include the plan to cover those so-called “missing middle” between the poor, who are covered by HEF, and those in formal employment, who are covered by NSSF. The plan will also cover the senior population not protected by the pension. This population accounted for 90% of those aged 60 years or older. In the manuscript (lines 478-492), we changed to: “The HEF is an important pillar of the Cambodian government's social security system, and our findings suggest that HEF membership should be expanded to cover people with chronic conditions. The Cambodian government recognized that the current social protection

	system has not yet covered those so-called “missing middle” between the poor, who are covered by the HEF, and those in formal employment, who are covered by the NSSF. Therefore, a new social health protection scheme targeting those in the informal economy and senior population without pensions, which accounts for 90% of people aged 60 years or older in Cambodia, should be created. This social protection scheme must go alongside with improving service coverage and quality. The success of such a model has been demonstrated by Thailand, a neighboring country of Cambodia.^{32 33} Thailand focused on improving public healthcare services and introduced three public health insurance schemes. One of them was the Universal Coverage Scheme, which covered 75% of the Thai population.³² Such a model might be too ambitious for Cambodia, since Thailand is more economically developed than Cambodia. However, this is still a model that Cambodia should be aiming for, so that quality health services for people with T2D and/or HTN will be more accessible”
d) In a period where GDP growth rates of most countries are far below, growing at 7% annually is a great opportunity to recommend to the government that it should invest in health and education. Particularly in health to create a real public system. Authors should provide an opinion about this in the discussion.	The annual growth of 7% might be somewhat misleading. Despite the economic growth looking great in percentages, the absolute term remains small (we have deleted this misleading sentence in the current version). Cambodia has spent more than its income in the past two decades. The deficit budget was filled with an international loan. However, the government has prioritized health and education in the last three decades together with other sectors such as agriculture and rural development, infrastructure, and governance. Within this context, we cannot recommend, based upon our findings, to invest more in health in general, but we do recommend expanding the T2D and HTN service and expanding the social protection system. Naturally, this also means investing more in health.

	Please see lines 478-492 and 462-467.
e) It is striking to see how an organizational improvement in the model of care (community based care model) would mean transferring added costs to the users. The public systems is acting as a private enterprise. Could authors provide an opinion about the risks of expanding new models that are more costly to the users.	This is indeed a striking finding, but also one which requires cautiousness in its interpretation. We acknowledge that this is one of our limitations that we could not state clearly whether the higher expenditure is from a community-based model or other factors because a large proportion of users used private services. The private sector is a significant potential confounding factor that we have limited knowledge about. We found the expenditure is higher, but we did not have the treatment outcome to cross-check that the higher cost leads to a better treatment outcome. We definitely need further research to have concrete evidence. We, therefore, suggest further studies to investigate the relationship between cost and treatment outcome. In the manuscript, we added “It is unclear what the influencing factors are because a large proportion of service users (80%) used private services in this OD.” (See line 497-499) We also added “... while drugs and consultation fee are not subsidized” to a sentence. The full sentence became “The higher unit costs in the community-based care were driven by adding PE components and field activities to the model while drugs and consultation fee are not subsidized.” We added, “We, therefore, cannot make a recommendation from this limited finding.” (See line 515-516) Finally, we removed “and expand community-based care for this population” from the recommendation in conclusion section.
f) Authors make no comparisons whatsoever	To address your comments above and below, we

with similar studies in other countries. Discussing with other countries realities may open up to new ways of interpreting results of the Cambodian reality.	add more literature (six papers) to help compare our findings with findings in other countries and make recommendations. This included (1) a series of four papers on the public-private mix published in the Lancet and (2) literature of social protection schemes and universal health coverage achievement in Thailand (two papers). Please see lines 453-460, lines 467-472, and lines 486-489.
g) Suggesting that a public private mix could be a next step forward without suggesting which could be specific characteristics of this mix leaves the readers with more questions than answers about how this could be a protective option for the population. International experiences on public private mix have shown a broad variety of results depending on how this is planned and implemented. Please comment about.	Regarding the public-private mix, we changed our recommendation and extensively revised the paragraph. Please see line 462-471. “To move forward in the direction of universal health coverage, meaning that people can access the health services they need without financial hardship, Cambodia should focus on expanding quality service coverage for people with T2D and/or HTN at public healthcare facilities across the country. Expanding quality services at public healthcare facilities may be the best suited approach to the Cambodian context, where dual practice is strong and regulation weak. The literature suggests that the public and private providers are not mutually exclusive and they shape each other's characteristics or sometimes so-called competition for health benefits.^{26 28-30} If public healthcare providers can provide quality services at affordable prices to the poor or those from low-income households, visits to private healthcare providers, who offer inferior services at higher prices, will decrease .^{26 28-30} The private healthcare providers will change their service provision to target the rich.^{26 28-30}” We removed the recommendation regarding the public-private mix from the conclusion. “However, these may require more time and resources. One potential strategy in the short run is to partner the private sector with the

public sector.”